# Nucleation of protein mesocrystals via oriented attachment

Alexander E. S. Van Driessche [1], Nani Van Gerven[2,3], Rick R. M. Joosten[4,5], Wai Li Ling [6], Maria Bacia[6], Nico Sommerdijk [7] & Mike Sleutel [2,3] ✉

Self-assembly of proteins holds great promise for the bottom-up design and production of synthetic biomaterials. In conventional approaches, designer proteins are pre-programmed with specific recognition sites that drive the association process towards a desired organized state. Although proven effective, this approach poses restrictions on the complexity and material properties of the end-state. An alternative, hierarchical approach that has found wide adoption for inorganic systems, relies on the production of crystalline nanoparticles that become the building blocks of a next-level assembly process driven by oriented attachment (OA). As it stands, OA has not yet been observed for protein systems. Here we employ cryo-transmission electron microscopy (cryoEM) in the high nucleation rate limit of protein crystals and map the self-assembly route at molecular resolution. We observe the initial formation of facetted nanocrystals that merge lattices by means of OA alignment well before contact is made, satisfying non-trivial symmetry rules in the process. As these nanocrystalline assemblies grow larger we witness imperfect docking events leading to oriented aggregation into mesocrystalline assemblies. These observations highlight the under-appreciated role of the interaction between crystalline nuclei, and the impact of OA on the crystallization process of proteins.

[1] Univ. Grenoble Alpes, CNRS, ISTerre, Grenoble, France. [2] Structural Biology Brussels, Vrije Universiteit Brussel, Brussels, Belgium. [3] Structural and Molecular Microbiology, Structural Biology Research Center, VIB, Brussels, Belgium. [4] Department of Chemical Engineering and Chemistry, Center of Multiscale Electron Microscopy, Eindhoven University of Technology, Eindhoven, The Netherlands. [5] Institute for Complex Molecular Systems, Eindhoven University of Technology, Eindhoven, The Netherlands. [6] Univ. Grenoble Alpes, CEA, CNRS, IRIG, IBS, Grenoble, France. [7] Department of Biochemistry, Radboud Institute of Molecular Life Sciences, Radboud University Medical Center, Geert Grooteplein, GA Nijmegen, The Netherlands. ✉email: mike.sleutel@vub.be

The simplest model of crystallization assumes that the seeds formed during the initial stages are structurally identical to the macroscopic crystals that spawn from these nuclei. This idea is the foundation of the classical nucleation theory (CNT)[1] and was until recently considered to be the most effective framework to describe nucleation[2]. This started to change when theoretical and experimental tools were developed that allowed scrutinizing the underlying assumptions of CNT[3]. Such enquiries to the nanoscopic mechanisms of nucleation have uncovered a broad range of nucleation pathways that do not fit the simple textbook CNT picture[4–6] which has led to the proposal of several non-classical models of nucleation that relax the structural restraints of the nucleus[7].

Of relevance to proteins as nucleating species is the so-called two-step nucleation model[8] where molecules first self-assemble into a metastable, liquid-like aggregate, which transforms into a crystalline cluster by a structural reorganization process. This model was initially formulated based on numerical simulation results[9], and later supported by (in)direct experimental evidence[10–13]. As it stands, two-step nucleation has emerged as the dominant model in the field[14,15], but that prepossession is unfounded for several reasons. First, there are recorded instances of crystal nucleation where protein molecules follow a nucleation pathway akin to the one laid out by CNT[5,16,17]. Secondly, the frequently proposed candidates for the non-crystalline precursor phase of the two-step model are submicron-sized particles that have been observed for numerous proteins[13,18–22]. The structural nature and composition, however, of these particles remain unknown. Their liquid-like nature has been suggested based on their propensity to flawlessly merge with the lattice of a mother crystal[11,18,23], but that argument goes by on the lessons learned from OA of inorganic nanocrystals[24]. Thirdly, the origin of the mesoscopic size of said particles is still poorly understood. There is an ongoing debate regarding the theoretical viability of the mechanism that stabilizes their size[25–28]. Even if one disregards these issues, the two-step model focuses narrowly on the initial stages of nucleation up until the formation of a crystalline cluster but makes no predictions regarding any later stages that may follow.

In this work, we address these unknowns by targeting a regime of nucleation where interactions between nuclei are more likely to occur. We work with glucose isomerase (GI) whose nucleation mechanism has been suggested to follow a two-step pathway[18], and for which groundwork on the characterization of the pre-nucleation particles has already been performed. Our in situ data for GI exposes crucial interactions between nuclei mediated by OA that determine the material properties of the final phase. These observations highlight the hitherto unknown role of the interaction between crystalline nuclei, and the impact of OA on the nucleation process of protein crystals.

## Results

**GI nanocrystals can merge lattices through OA**. We use cryoEM to follow the nucleation of a point mutant of GI (R387A) where the surface exposed residue arginine 387 is changed to an alanine. We show that R387A crystallizes in both the I222 and H32 space groups (Supplementary Fig. 1). In our optimized condition, H32 forms hundreds of microcrystals in a mother liquor volume of 10 µL within a timeframe of minutes. Such rapid nucleation increases the probability for nanocrystal interactions to occur in the solution. Almost immediately after mixing R387A with PEG 1000, we can resolve nanocrystalline particles (Fig. 1a, 1min40). These nanocrystals are facetted and their FFTs show clear diffraction patterns that match the predictions based on crystallographic data obtained from macroscopic crystals

(cryoEM: $6.7 \pm 0.1$ nm; standard deviation, $n = 33$; X-ray: $a = b = 6.64$ nm, 60°; Supplementary Fig. 2). On some occasions (9 out of 33 analyzed nanocrystals), we measure minor stretching of these intermolecular distances (between 6.8 and 6.9 nm) indicating that there is some flexibility in the nanocrystal lattices. We also point out that the facets are surprisingly smooth (straight), suggesting that they represent a Wulff shape that emerges out of the anisotropy in the surface tension. Moreover, the hexagonal shape of the nanocrystals is in line with what can be expected from a simple periodic bond chain analysis for the H32 space group.

On some occasions, rough crystal habits are also observed. More specifically, out of 118 analyzed nanocrystals, 24 had one or more rough facets, with an average Wenzel roughness of $1.1 \pm 0.3$ (standard deviation, $n = 24$). Such irregular boundaries are typically the result of kinetic coarsening at high supersaturation but they could also have formed via a non-classical nucleation process (e.g., the emergence of crystalline order from within a disordered GI cluster). Such a two-step scenario can be ruled out based on the absence of any disordered, liquid-like clusters in our cryoEM images. We conclude therefore that these initial GI nanocrystals nucleate via a one-step mechanism. The rough facets are likely remnants of the irregular shapes sampled by the crystalline clusters as they transition from sub- to super-critical dimensions. During the growth process that follows it is expected that they minimize the dangling bond density at their perimeter and adopt the rhombic Wulff shape in the process. This interpretation is reinforced by our observation of very small (<10 molecules per facet edge) rhombic nanocrystals that have sharp vertices.

Although most crystals are oriented parallel with their (001) face with respect to the imaging plane, some side views can also be discerned (Fig. 1b and Supplementary Fig. 3). The number of lattice planes visible along this direction varied between 3 and 10 molecular rows, demonstrating that these are three-dimensional crystals. From this observation, we can determine the lattice spacing along the $c$ axis, i.e., $8.3 \pm 0.1$ nm (standard deviation, $n = 10$) which is 5.7% larger than the 7.83 nm $c$-spacing derived from X-ray diffraction (Supplementary Table 2). To put these measurements in perspective, we looked at the typical variation in the lattice cell parameters for the I222 space group of GI based on depositions in the protein databank (www.rcsb.org). Based on 90 different entries, we arrive at a lattice variability of 2.8%. To account for the remaining difference, we hypothesize that the early formed nano-crystallites gradually compact to a more condensed state as they mature to larger sizes.

At later time points of the crystallization reaction (>3 min), we find groupings of similar nanocrystals that have merged into a unified lattice with no discernable stacking faults at their junctions (Fig. 1c–e). The crystal in Fig. 1c is of interest because it shows two smaller domains (1 and 2) that are in perfect co-alignment, and that make loose contact with a larger, third domain residing at a 4.3° angle. It is our assertion that such conglomerate structures are formed through an OA process. In fact, close inspection of the junction area between domains 2 and 3 reveals that both regions are in near contact with each other (Fig. 1d). It is tempting to speculate that the angular offset between these domains would eventually be eliminated by rotational diffusion, leading to the melding of both lattices once alignment had been reached[29]. Indeed, one such event is shown in Fig. 1e (and Supplementary Fig. 4), where we discern two nanocrystals that have melded in co-alignment into one unified lattice, held together by a joint molecular column at their interface (black arrow Fig. 1e and white arrow Supplementary Fig. 4c). Next, we measured the surface area of isolated nanocrystals and of post-docking nanocrystal assemblies (Fig. 1f).

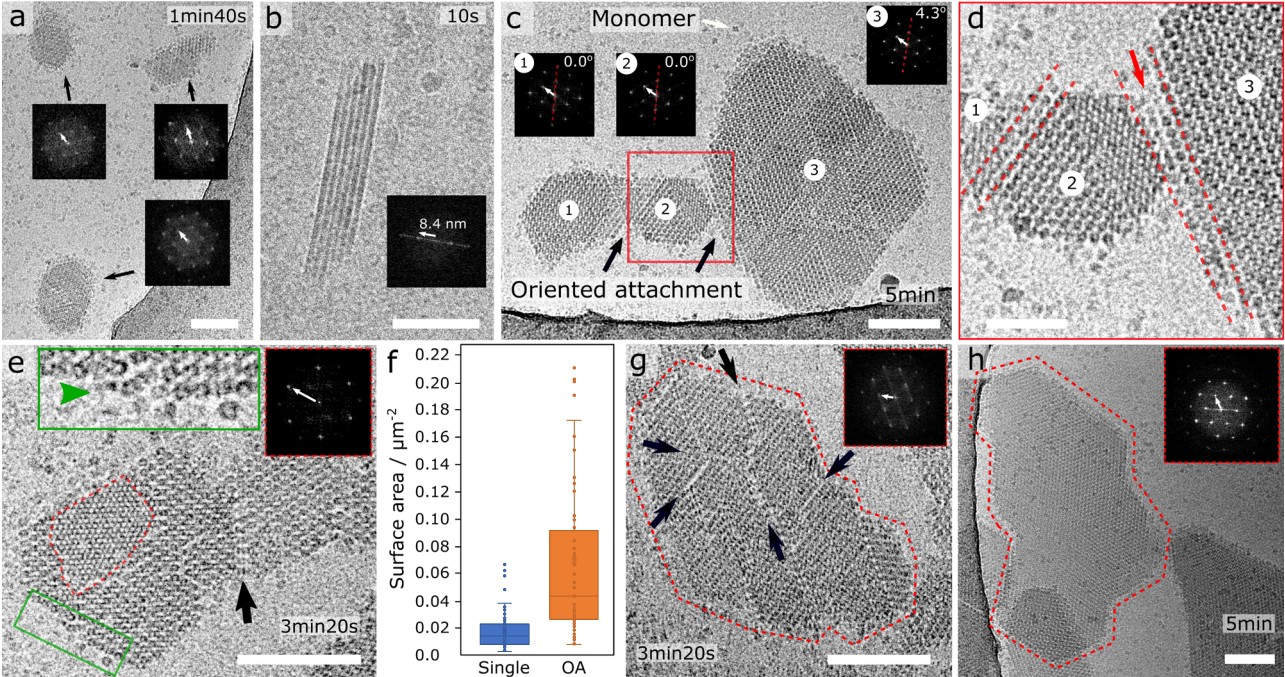

**Fig. 1 Oriented attachment of GI R387A nanocrystals. a** Submicron nanocrystals formed 1 min 40 s after mixing protein and precipitant; corresponding FFT images exhibit sharp maxima; **b** zoom-in of a particle rotated 90° with the c-axis in the plane of imaging (i.e. side view), showing 7 molecular rows with a lattice spacing along the c-axis of 8.4 nm; **c** oriented attachment of individually nucleated nanocrystals into a larger, merged lattice composed of domains 1 and 2, **d** making loose lateral contact with domain 3 at an angle of 4.3°; **e** two nanocrystals with merged lattices; red inset: FFT of the second molecular layer that is forming on the parent crystal; green inset: zoom-in of the unfinished molecular layer resolving local disorder and incoming growth units; **f** surface area of single ($n = 60$) and docked ($n = 61$) nanocrystals: center line, mean; box limits, upper and lower quartiles; whiskers, 1.5x interquartile range; points, outliers (**g**) and (**h**) large aligned nanocrystal assemblies with and without fault lines between the separate domain, respectively. Scalebar is 100 nm in panels **a**, **b**, **c**, **d**, **e**, **g** and **h**, and 50 nm in panel **d**. White arrows in the FFT's correspond to a resolution of 6.6nm unless stated otherwise.

On average, the latter is 2 to 4 times larger than the former. This result has two important implications: (i) larger structures are likely formed by multiple, consecutive docking events, and (ii) growth via crystal coalescence outpaces classical growth via single-molecule adsorption events. This means that this route of crystallization represents a kinetic shortcut towards the end-state.

A mechanism of cooperative, and guided merging of separately nucleated lattices is further implied by the observations of large (~0.5 μm) more erratically shaped structures that deviate from the expected Wulff shape and which display faint internal contrast lines that outline the individual rhombic clusters residing within the larger structure (black arrows in Fig. 1g). These lines are either grain boundaries (GB) and therefore represent local points of failure to completely merge lattices, or they are thin layers of solvent that still need to be expelled for contact to be completed. It is difficult to discern between the two possibilities at this stage, but successful docking of multiple nanocrystals is certainly possible as evidenced by the complex morphology in Fig. 1f that does not appear to have any discernable GBs.

We find even larger (>1 μm) composite structures with pronounced fault lines that separate homogeneous lattice domains (Fig. 2a–d). What is striking is that all these domains are in near-perfect alignment, which means that these super-structures are mesocrystals. This can be inferred from the FFT (Fig. 2b) of the highlighted region in Fig. 2a showing only minimal spread of the diffraction peaks. In panel c of Fig. 2, we also show the individual FFTs of regions 1 to 6, highlighting the various degrees of rotation. The mesocrystal in Fig. 2d exhibits even better alignment of its subdomains as demonstrated by Fig. 2e. Although, careful analysis of the orientations of domains

1, 2, and 3 reveals that they reside at angles $0 \pm 0.4°$, $-0.8 \pm 0.1°$, and $2 \pm 0.3°$ of each other taking domain 1 as a reference. The typical area of the individual domains within the mesocrystals is of the order of 0.15 μm², i.e., comparable to the surface area of the nanocrystal assemblies where lattice docking was successful (see "OA" in Fig. 1f). Moreover, the distance across the GBs is in the range of one or two molecular rows, suggesting that the shape complementarity of the various domains is the result of the addition of new GI molecules at the imperfect contact areas that form after docking (Fig. 2f).

We do, however, also find examples where the inter-crystal alignment has either failed or has not yet reached completion before cryo-quenching took place, resulting in an overall lack of long-range order (Fig. 3). We identify an absence of lattice alignment between nanocrystals that interact laterally or stack axially (Fig. 3a–c). For example, the green area in Fig. 3b highlights two stacked domains exhibiting registry between their corresponding lattices, but clear misalignment with the region enclosed in red. Fig. 3c is an example of a lack of axial alignment where the FFT of the interlaced pattern (green) reveals two independent lattices residing at an angle of $25 \pm 0.5°$. Although cryoEM images represent single snapshots in time, it is tantalizing to speculate that these structures may eventually relax into a lower energy state possessing more long-range order. Such gradual orientational ordering has indeed been observed in other colloidal systems that are vertically stacked[30] and may well exist here too.

There may, however, be a limit to this self-organization process. It seems unlikely that the disordered grouping of over 100 nanocrystals shown in Fig. 3d will transform into a single, unified (meso)crystal. Indeed, light microscopy observations of

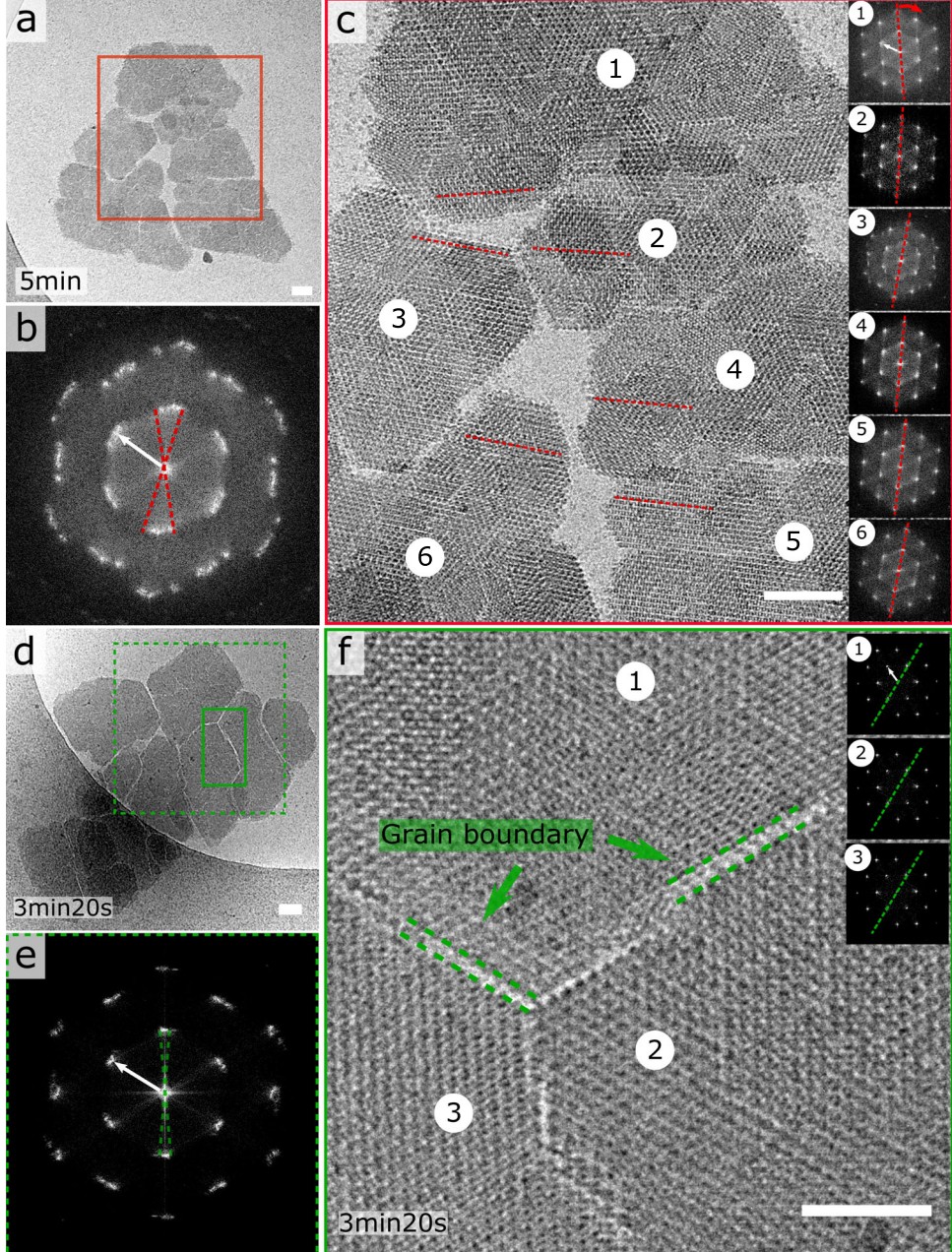

**Fig. 2 GI R387A mesocrystals. a**, **c**, **d** Large micron-sized composite nanocrystal structures with pronounced fault lines that separate lattice domains that are in near-alignment as demonstrated by their respective FFTs (**b**, **e**); zoom-in of the grain boundaries that span one or two molecular distances between the individual domains (**f**). The inset in panel (**f**) demonstrates the high degree of lattice order within each separate domain: domain 2 shown as representative example. Scalebar is 50 nm in panels **a**, **c** and **d**, and 25 nm in panel (**f**). White arrows in the FFT's correspond to a resolution of 6.6 nm.

the earliest observable crystals appear twinned and full of defects (Supplementary Fig. 5) reminiscent of the alignment failures we see at the nanoscopic level. We do point out that a full reconstruction of all pathways and their associated throughput is not currently feasible using our cryoEM approach. At the latest stages of the assembly process, the particles become (prohibitively) large for a meaningful cryoEM characterization because: (i) the blotting process may introduce a bias towards smaller particles by filtering out larger ones, and (ii) such thicker objects become opaque to the electron beam. We expect that a combination of blotting-free grid preparation protocols[31] and sectioning techniques can help to expand the experimental window on a range of self-assembly processes in the future. It is

also worth considering whether the adopted cryoEM methodology in which a 3 μL aliquot of the reaction volume is applied onto a cryoEM grid and blotted, accurately captures the nucleation process that takes place in the liquid bulk. One could argue that the most likely source of discrepancies between the imaged state and the state in the bulk is the blotting process which will bias the EM sample towards smaller particles (as discussed above), and create solutal flows that introduce additional shear forces. Having said that, we do see a clear temporal trend that starts from the emergence of isolated smaller crystals, to larger crystals with distinct GBs and ultimately to mesocrystals. Such a trend can only have emerged in the total reaction volume and is not an artifact of the grid preparation protocol.

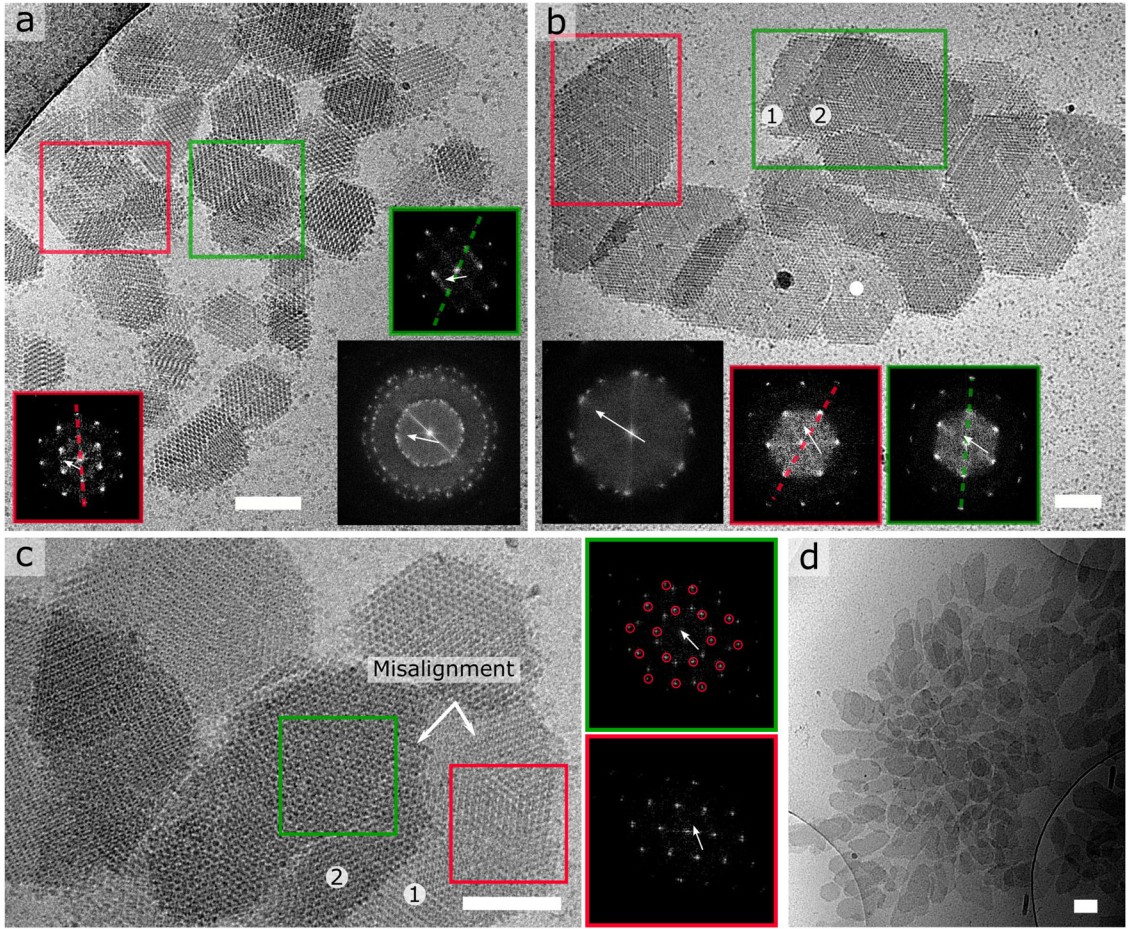

**Fig. 3 Lack of inter-crystal alignment in large nanocrystal assemblies. a** Poly-crystalline cluster with local hotspots of alignment (see FFT insets), **b** example of two vertically stacked domains exhibiting registry between their corresponding lattices (blue) but a 26° misalignment with the region enclosed in red, **c** example of lack of axial alignment between two nanocrystals where the FFT of the interlaced pattern (green) reveals two independent lattices residing at an angle of 25°, **d** disordered grouping of over 100 nanocrystals. Scalebar is 100 nm. White arrows in the FFT's correspond to a resolution of 6.6 nm.

**Rotational symmetry restraints limit the probability of success for the final jump to contact.** The OA-mediated accretion of R387A nanocrystals is remarkable considering the non-trivial symmetry requirements of the H32 space group (Supplementary Table 1 and Supplementary Discussion). Based on the crystal structure of the planar H32 crystals observed in cryoEM, we identify the (001) plane as the dominant orientation. The 3-fold screw axis is perpendicular to the (001) plane resulting in 3 distinct GI orientations (Fig. 4: designated as 1, 2, and 3; Supplementary Fig. 6). The nature of the crystallographic symmetry is such that identically oriented GI molecules do not engage in lattice contacts. This constraint poses additional registry requirements for OA to be successful, as highlighted in Fig. 4. Even if two proximate nanocrystals are in rotational alignment, approximately only a third of all configurations will lead to proper docking of both lattices (Supplementary Fig. 7). For those scenarios where the bond formation is not possible, crystals will either need to slide laterally along the interface or diffuse away to reinitialize altogether. We use PEG 1000 as a crystallization agent that induces an attractive depletion force that scales with the size of the interface between crystallites as they approach each other[32]. Hence, for larger clusters, it may become increasingly unlikely to find the proper translational register as a result of the depletion force even though rotational alignment has already been reached. This hypothesis fits with the cryoEM observations, which show

that GBs tend to develop within larger conglomerates resulting in the formation of mesocrystals.

## Discussion
Macromolecular crystallization is an astounding feat of nature. Even though proteins are large, dynamic, and often highly anisotropic molecules, they can form a minimal assembly that guides incoming molecules to the registry as instructed by its internal rules of symmetry. Understanding how this nucleus form has been the subject of debate for over two decades now. The two-step nucleation model is arguably the most popular and is often considered as a consensus view on this subject. Direct experimental evidence for a nucleation trajectory akin to the predictions of the two-step model have recently been provided by Houben et al.[10] for the case of ferritin. But the process that they witness is more nuanced than the original two-step scheme. They record the initial formation of disordered ferritin aggregates that tend to increase in both order and density from their surface towards their interior in a cooperative process of gradual desolvation. The process of self-assembly that they describe is in stark contrast with our observations here for GI R387A. Although there have been indications that GI may also first condense into a non-crystalline precursor under certain conditions[18], our cryoEM observations identify the earliest assemblies as nanoscopic renditions of the macroscopic crystals that emerge at the end of the

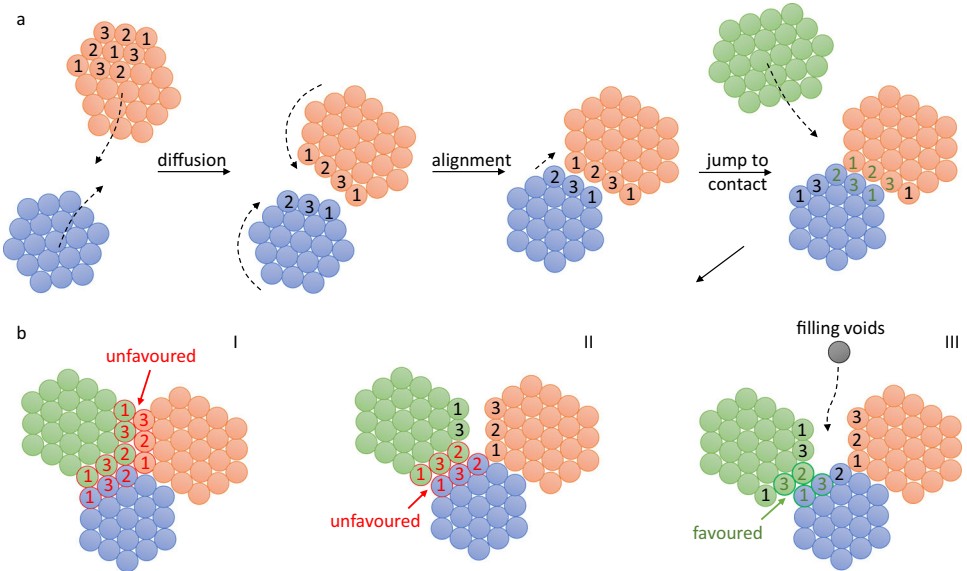

**Fig. 4 Model for OA of GI nanocrystals.** GI molecules pack along a 3-fold axis within the (001) plane in which we discern three different GI orientations (1,2,3). Nearest neighbors exclude GI molecules with identical orientations; **a** Simplified scheme of self-assembly: freely diffusing nanocrystals approach each other, followed by rotational and translational adjustments to align both lattices. Alignment facilitates a final jump to contact by desolvation of the surface patches that partake in lattice contact formation; **b** Illustration of three different scenarios for further growth: I and II violate H32 symmetry rules and are likely to lead to the formation of a GB at the interface; III leads to successful merger of all three lattices and the resulting voids can be filled by monomer addition.

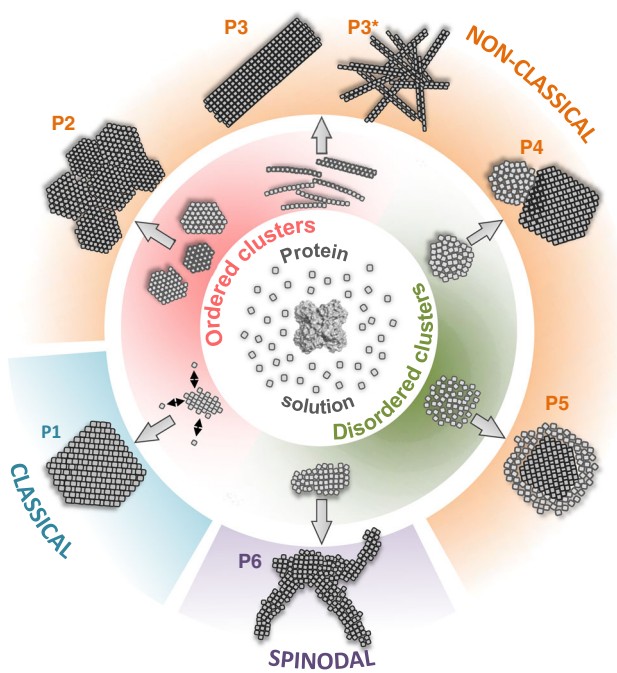

**Fig. 5 Protein crystal nucleation pathways that have been experimentally observed at the nanoscale.** P1: one-step nucleation solely involving crystalline clusters throughout the entire pathway (glucose isomerase I222[5]); P2 and P3: involving oriented attachment of 2D, 3D and 1D crystalline clusters into larger ordered assemblies (glucose isomerase H32 and P2 2 2[5]); P3*: spinodal decomposition limit of the P3 scenario leading to kinetic jamming (gel)[5]; P4: self-seeded nucleation of crystalline clusters on the surfaces of solid, amorphous condensates (lysozyme[6]); P5: two-step nucleation comprising initial densification into loose disordered clusters, followed by gradual local desolvation and densification into a crystalline array (ferritin[10]); P6: aggregation in the high supersaturation limit with poorly ordered clusters[5].

crystallization process. These observations essentially mean that GI H32 crystallites nucleate in qualitative accordance with the CNT model. Based on our previous work for GI[5,16] and the work of Houben et al.[10], a picture emerges where proteins can nucleate through multiple routes that are conceptually diverse and which do not fit our idealized views (Fig. 5).

The differentiation between one or two-step nucleation becomes less relevant for situations where the number of nucleation centers is relatively high, such that interactions between nuclei cannot be disregarded. R387A showcases such a regime of nucleation in the high concentration limit of nucleating entities. Here we see the clear interplay between independently formed clusters in a manner reminiscent to a host of inorganic systems whereby nuclei diffuse in solution, collide, and coalesce to form larger unified structures and mesocrystals[33]. Interactions between nuclei are by no means non-classical and can be traced back to Smoluchowski's treatment of coagulation in 1916[34]. What might be considered as non-classical is the manner how these clusters interact. More specifically, the merger of lattices in a defect-free manner is unlikely to occur simply by chance. Rather, nanoparticles must experience a torque that guides each other towards (near) perfect registry[29]. That directed maneuvering has long been recognized as a widespread mechanism of nanoparticle assembly and is referred to as OA[35]. The steering torque that guides OA can often be attributed to the dipole moment of the nanocrystals[36]. This is not the case for GI because the net dipole moment of a GI tetramer, i.e., the crystals' building block, is zero because the dipole moments of the monomers cancel each other. And yet, the existence of GI mesocrystals is a strong indication that such a guiding mechanism exists because we see the alignment of crystalline domains that do not have any bridging contact points between their respective lattices. This demonstrates that alignment occurs before docking, i.e., OA is facilitated by lattice alignment, and presumably at a relatively long-range (~10 nm). Oriented attachment for dipole-free systems has been attributed to short-range Van der Waals attractions, but Liu et al. have recently suggested that local water structuring could play a role in

the long-range steering that takes place before the jump to contact[37]. The final step will entail the desolvation of the respective surfaces that become buried at the docking interface. This process is still far from being understood, but recent molecular dynamics simulations[38] suggest that the structure and fluctuations of the hydration shell may have an important impact on the height of the activation barrier for desolvation, and by extension the net rate at which docking is expected to take place.

At the same time, R387A also demonstrates the limits of OA. Initially, smaller crystallites undergo near-perfect OA into a unified structure with little or no defects at their junctions. But as these structures grow larger and develop more complex shapes, docking of lattices becomes hampered, GB appear and mosaicity increases resulting in a colloidal growth process[39] that generates mesocrystals. This transition from an initial stage characterized by oriented attachment into unified lattices towards the second stage of oriented aggregation of crystalline domains correlates with the typical size of the building blocks ($\pm 0.15 m^2$). Understanding the underlying mechanims that determine this transition could aid in the design of novel biomaterials leveraging precise control over the self-assembly mechanism to tune the size, aspect ratio[40], and polycrystallinity of the final phase. Moreover, with the cryoEM revolution in structural biology that is focusing more on electron diffraction[41], combined with the need for protein nanocrystals for XFEL diffraction[42], we believe a better understanding of macromolecular OA could contribute in these research domains as well.

## Methods

**Protein production and purification**. GI R387A was recombinantly expressed in *E. coli* BL21(DE3) after induction at $OD_{600nm}$ of 0.7 with 1 mM IPTG for 3 h at 37 °C. Cells were harvested by centrifugation at 6238 g for 15 min and resuspended in 100 mM Tris-HCl pH 7.3, 1 mM ethylenediaminetetraacetic acid (4 mL g$^{-1}$ wet cells) supplemented with 5 μM leupeptin, 1 mM 4-(2-aminoethyl)benzenesulfonyl fluoride (AEBSF), 100 μg mL$^{-1}$ lysozyme and 20 μg mL$^{-1}$ DNase I and incubated for 30 min at 4 °C. Subsequently, MgCl$_2$ was added to a final concentration of 10 mM, and cells were lysed by two passages in a Constant System Cell Cracker at 20 kpsi at 4 °C and cell debris was removed by centrifugation at 48,400 × g for 45 min at 4 °C. The cytoplasmic extract was incubated for 10 min at 65 °C and the insoluble fraction was removed by centrifugation at 48,400 × g for 45 min at 4 °C. The supernatant was filtrated through a 0.22 μm pore filter and loaded on a 5 mL pre-packed Hitrap Q FF column (GE Healthcare) equilibrated with buffer A (50 mM bis-tris-HCl pH 6.0, 10 mM NaCl). The column was then washed with 40 bed volumes of 20% buffer B (50 mM bis-tris-HCl pH 6.0, 1 M NaCl) and bound proteins were eluted with a linear gradient of 20–50% buffer B over 10 bed volumes. Fractions containing R387A, as determined by SDS–PAGE, were pooled and supplemented with ammonium sulfate to a final concentration of 1.5 M and loaded on a 5 mL pre-packed HiTrap Phenyl HP column (GE Healthcare) equilibrated with buffer A (100 mM Tris pH 7.3, 1.5 M ammonium sulfate). The column was then washed with 40 bed volumes of 25% buffer B (100 mM Tris pH 7.3) and bound proteins were eluted with a linear gradient of 25–85% buffer B over 15 bed volumes. Fractions containing R387A were pooled and dialyzed (Spectra/Por Standard RC Turbing: 12–14 kDa; Spectrumlabs) against 10 mM Hepes 7.0, 1 mM MgCl$_2$ overnight at 4 °C (buffer was replaced twice) and concentrated in a 100 kDa molecular weight cutoff spin concentrator (Amicon Ultra −15 Cellulose, Millipore) to a typical final concentration of 30 mg mL$^{-1}$.

**Glucose isomerase crystallization**. To trigger crystallization of R387A, the protein stock solutions were mixed at 22 °C with an equal volume of 100 mM Hepes 7.0, 200 mM MgCl$_2$ and 8% (w/v) PEG$_{1000}$.

**Cryo-transmission electron microscopy**. For cryoEM, 200 mesh Cu grids with Quantifoil R 2/2 holey carbon films (Quantifoil Micro Tools GmbH) were used. Sample preparation was performed using an automated vitrification robot (FEI Vitrobot Mark III) for plunging in liquid ethane cooled by liquid nitrogen[43]. All electron microscopy grids were surface plasma treated for 40 s using a Cressington 208 carbon coater prior to use. A few microliters of protein solution were applied to the treated grid in the humidity and temperature-controlled chamber of the Vitrobot. After automatic blotting, the grid was plunged into the liquid ethane and vitrified at a cooling rate > 10$^4$ K/s so that the sample is preserved in amorphous ice. We choose $t_0$ as the moment where we induce supersaturation with respect to the crystalline phase (i.e., mixing of the protein with the precipitant solution) and

$t_{end}$ as the time at which crystals become detectable using light microscopy. The exact time point of the samples as indicated in the main text is defined as the moment (after blotting excess liquid) when the electron microscopy grid is plunged into the liquid ethane. The samples were imaged with the TU/e cryoTITAN (FEI, www.cryotem.nl) operated at 300 kV, equipped with a field emission gun (FEG), a post-column Gatan Energy Filter (GIF), and a post-GIF 2k × 2k Gatan CCD camera. Images were acquired in low-dose mode at a magnification of either 24,000× with a nominal defocus of −5 μm or 11,500× with −10 μm defocus. During low-dose mode imaging, the search of relevant areas on the grids is done at low magnification, and focusing is always performed in areas distinct from the areas of interest. The area of interest is only exposed at the desired data collection magnification during image capture to minimize radiation exposure (and thus possible damage).

**Crystallographic analysis**. The nearest crystallographic neighbors of the GI molecule are generated using Chimera 1.13.1. Residues partaking in lattice contacts are identified by calculating the accessible surface area (ASA) on a per-residue level using AREAIMOL of the CCP4 software suite[44]. ASAs are determined for both the starting models as well as the models consisting of the GI molecule and its nearest neighbor using a probe radius of 1.4 Å. Residues with a non-zero difference in accessible surface area (ΔASA) are (partially) buried in the bound complex and therefore considered to be part of the lattice contact patch. Hydrogen bond pairs are identified using the *FindHBond* tool in Chimera 1.13.1rc using default settings, and salt-bridges are identified using the PDBePISA (http://www.ebi.ac.uk/pdbe/pisa/) and the 2P2I (http://2p2idb.cnrs-mrs.fr/2p2i_inspector.html) protein interaction webservers.

**Roughness metric**. As a measure of facet roughness, we use the Wenzel roughness ($R_W$) by calculating the ratio of the circumference of the nanocrystals and the theoretical, geometric circumference of a similarly sized crystal with straight facets[45]. For crystals with perfectly smooth facets, $R_W$ is expected to be 1.0.

**Statistics and reproducibility**. Two separate cryoEM data sets of two separate recombinant protein preparations were collected and yielded reproducible results. The images used in the figure panels were taken from both data sets.

**Reporting summary**. Further information on research design is available in the Nature Research Reporting Summary linked to this article.

## Data availability

The data that support the findings of this study are available from the corresponding author upon reasonable request. The glucose isomerase S171W structure can be found with the accession code 7BJZ at rcsb.org.

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

## Acknowledgements

We thank Dr. Guy Schoehn for his indispensable support with cryoEM data collection and insightful discussions. M.S. acknowledges financial support by the FWO under projects G0H5316N and 1516215N. This work used the platforms of the Grenoble Instruct-ERIC center (ISBG; UAR 3518 CNRS-CEA-UGA-EMBL) within the Grenoble Partnership for Structural Biology (PSB), supported by FRISBI (ANR-10-INBS-0005-02) and GRAL, financed within the University Grenoble Alpes graduate school (Ecoles Universitaires de Recherche) CBH-EUR-GS (ANR-17-EURE-0003). The electron microscope facility is supported by the Auvergne-Rhône-Alpes Region, the Fondation Recherche Medicale (FRM), the fonds FEDER and the GIS-Infrastructures en Biologie Sante et Agronomie (IBiSA). IBS acknowledges integration into the Interdisciplinary Research Institute of Grenoble (IRIG, CEA).

## Author contributions

M.S. and A.E.S.V.D. designed the project and carried out the crystallization and data processing. N.V.G. cloned, recombinantly expressed, and purified R387A. Cryogenic freezing and cryoEM imaging were performed by W.L.L., M.B. and R.R.M.J. MS supervised the study. M.S., A.E.S.V.D. and N.S. wrote the paper, with contributions from all authors.

## Competing interests

The authors declare no competing interests.
