## [Peer Review File · Nature Communications]

REVIEWER COMMENTS

Reviewer #1 (Remarks to the Author):

The describes important observations related to OA in protein crystallization. The presented evidence is sound. OA is a critical process whose understanding is only starting to emerge.

There are several points to address before the work can be published:

1. The initially formed small crystals have ill-defined boundaries (no facets, Fig. 1a). It may imply nonclassical nucleation/growth. Must be addressed in the text.
2. Eventually, OA leads to large crystals, as one can imply from the data. However, mesocrystals often seem as a product rather than intermediate in the system, according to the text that focuses on grain boundaries. The percentage of mismatched mesocrystals vs. OA leading to large crystals must be clarified (quantitatively, if possible). In addition, thin cryo-TEM samples discriminate against larger particles that may be present and show additional modes of OA in 3D. These particles may be lost due to blotting. It should be clarified.
3. Desolvation should play an important role in OA. It is briefly mentioned, but needs to be addressed more explicitly.
4. A statement on stability of the cryo samples under the imaging conditions (low electron dose?) should be given in the article text before presenting the EM data.
5. The cartoon in Fig. 4 is not very informative.
6. I find the title misleading. Nonclassical nucleation mixed with OA makes it vague. I would focus on OA in the title.
7. Scale bars in FFTs must be given

Reviewer #2 (Remarks to the Author):

The manuscript by van Driessche et al. discusses

"Nonclassical nucleation of protein mesocrystals via oriented attachment (OA)"

based on cryoEM data on Gl.

The topic is of interest in general, but the authors should address the following points:

Generally, a more profound discussion on the comparison with colloids would be desirable.

Also, there does not seem to be a quantitative kinetic plot such as an observable as a function of time.

The work is not as quantitative as would be desirable.

Formal issues should be checked, such as language (e.g. line 195 "it's" should be "its") and inconsistencies in the references (including the authors' own names).

Introduction

line 52

"As it stands, two-step nucleation has emerged as the consensus in the field, but this fact remains difficult to reconcile with several issues."

Are the authors implying that it is really consensus for every type of system in this field ?

line 87

"We also point out that the facets, although rough on some occasions, are surprisingly smooth, suggesting that they represent a Wulff shape that emerges out of the anisotropy in the

surface tension."

It would improve the quality of the manuscript substantially if the authors could quantify this (how often is "some occasions" ? how rough ?)

Dimensionality issues (2D vs 3D and the differences between these; consequences of these)

- is this actually a (quasi)2D study ?
- how relevant are (quasi)2D studies to draw conclusions on 3D protein crystals which are used for structure determination ?
- interface effects (or from the limited reaction volume compared to "normal" bulk crystallization) ?
- the nanocrystal monolayers are apparently aligned perfectly flat at the surface, none is tilted ?

How about other orientations ? Can they be excluded ? Is this representative of bulk crystallization ?

- the behavior in the z dimension remains unclear.

Is there a "z-stacking" as observed for other pre-oriented building blocks (e.g. Ou et al., 2020, Kinetic pathways of crystallization at the nanoscale)?

Can the monolayers diffuse on top of each other?

It seems that the monolayers merge first before a new layer grows on top.

How does the new monolayer then grow?

Is it a huge monolayer or several smaller ones diffusing on top?

Or do only monomers attach in z-direction?

Reviewer #3 (Remarks to the Author):

This paper describes the nucleation behaviour of the protein glucose isomerase in the high rate regime where interactions between the formed particles cannot be ignored. The used technique, CryoEM, allows to determine molecular-scale time-stills during the nucleation process upon cryogenic freezing. The authors find that, rather than liquid-like, the formed nanoparticles have a crystalline nature showing a structure of the macroscopic crystals. This thus seems to follow a classical route, as has been shown for other protein systems. However and moreover, these nanoparticles undergo oriented attachment (OA) among each other to form composite structures with grain boundaries and even perfectly aligned constructs. It is the first time that OA, a mechanism identified for inorganic nanocrystals, is observed for a protein system. The authors speculate that this mechanism is a universal mechanism for proteins, given the right fast nucleation rate conditions. Indeed they show with this additional mechanism, that proteins can nucleate through diverse routes.

The paper is very convincing and the level of detail from the CryoEM technique is amazing. The paper is of very high importance to the community and outside it as it counters the two-step nucleation mechanism consensus that emerged in the community the last few years. The realization that each protein system picks, depending on the prevailing conditions, the least rate limiting route is interesting and leaves the question to what parameters determine that such a route is the fastest. It is therefore certainly a paper which I feel will have a high impact on the thinking in the field.

This realization of divers routes available to the protein to nucleate would be appealing to show in a schematic.

The CryoEM strongly relies on the instant cryo-freezing by plunging the sample in liquid ethane. The reader might not be up to date with this technique. It would be good to explain the technique, either in the methods section or in the supplementary information, directed towards, on the one hand, the influence of the fast but not instantaneous heat transfer during plunging and, on the other hand, the effect of the measurement itself on the quality of the frozen sample.

The section "OA may be a universal mechanism for proteins" lacks clarity. For instance, the remark on a host of enigmatic experimental observations (Page 5, line 156) is seemingly followed by a seemingly unrelated general statement. It is unclear whether the "this" in line 158 relates to the observations or the general sentence following it. This sentence furthermore identifies a further nucleation mechanism where the 2-step mechanism is combined with non-oriented attachment, perhaps a mechanism that has to be highlighted earlier. Then a discussion follows on the

intermediate species that may pre-exist in the used solutions, falling out of the blue sky. A reference to the reported seeding behaviour (line 171) is missing and it is actually unclear exactly what is meant with the term. Then, without further proof it is stated that the stock solution is slightly supersaturated (or is the red * in figure 5b indicating the solution conditions used?). It seems using supersaturated stock solution may cloud the interpretation of the observations as pre-existing clusters influence the nucleation behaviour. A growth rate is estimated (line 174) but it is unclear of what.

The authors clearly point out that OA seem rather unlikely as it needs the nanoparticles to experience a torque on a relatively large distance guiding the particles towards lattice alignment. They suggest that local water structure could play a role in this long range particle steering. Can the authors be a bit more specific?

The word "fact" in page 2 line 52 seems misplaced as after it reasons are given for the 2 step nucleation consensus not being a fact.

Page 3, line 79/80: high nucleation rates: how high? Compared to what?

Page 3 line 90 and further, figure 1. The times in text and figure are not consistent.

The last sentence of the introduction states OA to be a separate phenomenon from primary nucleation. This seems not consistent with the rest of the paper where it seems to be included in the nucleation process.

Page 5 line 147: there is no article reference listed for the attractive force between nanoparticles in the presence of polymers. What is actually the effect of PEG on the solubility of the protein? Would the occurrence of OA then be a complex function of the effect of the used PEG chain length PEG on the solubility and of the induced attraction due to the kind of PEG used?

The numbering of the figures is wrong at some places, for instance page 6, line 167, fig5b.

REVIEWER COMMENTS

We were truly delighted to receive such constructive feedback and insightful comments on our work. Based on our reading of the comments, we understood that the reviewers agree with the backbone of the manuscript, but that additional polishing was required before publication could be considered. Inspired by their suggestions we have made major revisions to the text, and we have gone to considerable lengths to collect additional experimental data that allows us to make more quantitative statements regarding the nucleation process. We provide an annotated version of the new manuscript to facilitate the tracking of the changes that we implemented. By integrating the suggested alterations to the manuscript we believe it has considerably improved. And for this we want to thank the reviewers once again. Below we reproduce the reviewer comments verbatim, interspersed with our detailed answers.

Reviewer #1 (Remarks to the Author):

The describes important observations related to OA in protein crystallization. The presented evidence is sound. OA is a critical process whose understanding is only starting to emerge. There are several points to address before the work can be published: 1. The initially formed small crystals have ill-defined boundaries (no facets, Fig. 1a). It may imply nonclassical nucleation/growth. Must be addressed in the text.

We apologize for not having addressed this already. The reviewer is correct in pointing out that some nanocrystals have ill-defined boundaries. Per the suggestion of Reviewer 2, we have quantified the roughness of crystal facets in more detail. 24% of the crystals that we analyzed had one or more rough facets. As a metric of roughness, we determined the Wenzel parameter (R_W is 1 for a straight boundary; cfr. the full answer to Rev. 2 for more details). The average R_W of the crystal perimeters that we analyzed was 1.1 ± 0.3 , ranging from 1 to 1.3.

Ill-defined boundaries could either have formed via a non-classical nucleation process (e.g. emergence of crystalline order within a disordered array), but it is also possible that they have classical origins (e.g. kinetic roughening at high supersaturation; irregular but crystalline nucleus shape). For the former, we should be able to detect the amorphous precursor species to faceted nanocrystals. To explore this hypothesis further, we have collected additional data at short incubation times (1min). We find no examples of any disordered, liquid-like clusters that predate the nanocrystals. We conclude that the GI nanocrystals nucleate via classical mechanisms, i.e. lattice formation occurs contemporaneous with densification. This conclusion is reinforced by our observation of extremely small nanocrystals with clear facets (see example below with the dashed white line serving as a guide for the eye; expanded for clarity). Rather than invoking non-classical arguments, we believe that rough facets are the result of kinetic coarsening: newly formed nuclei will not necessarily have a rhombic habit defined by straight edges and sharp vertices. Incomplete, rough facets will tend to be short-lived due to the high kink density (and associated high surface energy), and will quickly adopt the rhombic Wulff shape that we see for most larger crystallites. These arguments have been succinctly incorporated into the main text.

2. Eventually, OA leads to large crystals, as one can imply from the data. However, mesocrystals often seem as a product rather than intermediate in the system, according to the text that focuses on grain boundaries. The percentage of mismatched mesocrystals vs. OA leading to large crystals must be clarified (quantitatively, if possible). In addition, thin cryo-TEM samples discriminate against larger particles that may be present and show additional modes of OA in 3D. These particles may be lost due to blotting. It should be clarified.

The suggested statistics would provide us with valuable insights into the efficiency and associated success rate of the OA process. Unfortunately, however, that level of quantitative data is not accessible from our experimental data. Our cryoEM images of plunge-frozen aliquots of the reaction mixture provide still snapshots at different points in time of the reaction. We cannot, however, determine which objects (i.e. OA nanocrystals, mesocrystals) eventually grow out to become large crystals. We cannot exclude the possibility that even heavily mismatched mesocrystals will serve as the nucleation center of a macroscopic crystal. For this, we would need to be able to image later stages of the crystallization process. But, as rightfully pointed out by the reviewer, that is not possible for two reasons: (i) potential blotting bias, and (ii) micron sized crystals appear as opaque objects in the cryoEM images, i.e. their thickness does not allow us to resolve the growth center. The latter would be required for us to know whether any lattice-mismatches (as a result of imperfect OA) still reside in the nucleation cores of larger crystals.

The fact remains that our experimental limitations create a blind spot towards later stages of the crystallization process. Per the reviewers' suggestion and in the interest of full disclosure to the reader we have included a paragraph on these limitations:

"We do point that a full reconstruction of all pathways and their associated throughput is not currently feasible using our cryoEM approach. At the latest stages of the assembly process, the particles become (prohibitively) large for a meaningful cryoEM characterization: (i) the blotting process may introduce a bias towards filtering out larger particles, and (ii) such thicker objects become opaque towards the electron beam. We expect that a combination of blotting-free grid preparation protocols and sectioning techniques can help expand the experimental window on a range of self-assembly processes in the future."

3. Desolvation should play an important role in OA. It is briefly mentioned, but needs to be addressed more explicitly.

Desolvation is indeed expected to play an important role in OA as it is a crucial step in lattice bond formation. Our data suggests that nanocrystals can become aligned before contact is made – desolvation is therefore expected to only impact the latest stage of the OA process, i.e. the jump to contact. To gain more insight into the final stages of OA for GI, we would require time-resolved data (e.g. via *in situ* TEM). Time-resolved imaging that is combined with (near)-molecular resolution is, however, not yet experimentally attainable (predominantly due to radiation damage). Having said that, we did incorporate a brief paragraph in the discussion section on the relevance of desolvation for OA and how it might impact our system:

“That final step will entail the desolvation of the respective surfaces that become buried at the docking interface. This process is still far from being fully understood, but recent molecular dynamics simulations suggest that the structure of and fluctuations in the hydration shell may have an important impact on the height of the activation barrier for desolvation, and by extension the net rate at which docking is expected to take place.”

4. A statement on stability of the cryo samples under the imaging conditions (low electron dose?) should be given in the article text before presenting the EM data.

Indeed, all imaging was performed in low-dose mode, a standard imaging technique where the search is done at low magnification and focusing is done away from the area of interest. The latter is only exposed at the final magnification when the image is taken to minimize radiation damage. For our samples, we did not exceed 50 electrons/angstrom². Using this approach, we could confirm that the sample remained stable by collecting a second exposure which showed no obvious signs of radiation damage. Below, we present a side-by-side comparison of two consecutive exposures of an identical specimen area. Apart from a minor drift in the focal plane, we do not identify any clear signs of radiation damage, confirming that our low-dose imaging mode does not perturb the sample in a meaningful way. To be clear, we only used primary exposures to produce the figures of the main text and SI – secondary exposures have been used for housekeeping purposes only.

Following the suggestion of the reviewer, we have added a detailed explanation of our imaging strategy to the Mat&Met section of the manuscript.

5. The cartoon in Fig. 4 is not very informative.

We would suggest moving the cartoon to the SI document. This cartoon is intended for a non-specialist audience. It is meant to serve two purposes: (i) illustrate the general scheme of OA, and more importantly (ii) highlight the additional registry requirements for the H32 space group pertaining to the threefold screw axis.

6. I find the title misleading. Nonclassical nucleation mixed with OA makes it vague. I would focus on OA in the title.

Agreed. Most readers will associate “nonclassical” with a variant of “two-step”. We propose the following change to the title: “Nucleation of protein mesocrystals via oriented attachment”

7. Scale bars in FFTs must be given.

We have highlighted the radius that corresponds to the intermolecular distance within the (001) plane (i.e. 6.6nm). This effectively serves as a calibration of the FFT images.

Reviewer #2 (Remarks to the Author):

The manuscript by van Driessche et al. discusses "Nonclassical nucleation of protein mesocrystals via oriented attachment (OA)" based on cryoEM data on GI.

The topic is of interest in general, but the authors should address the following points:

Generally, a more profound discussion on the comparison with colloids would be desirable.

We had initially considered to broaden the discussion further and expand the comparisons to other colloidal systems. Having said that, after more than two decades of research on OA, the field has become fully matured, and the number of confirmed OA cases has become almost enumerable. We would prefer to reduce such comparisons to other systems to a minimum so as not to distract the reader too much from the research topic presented in this work. Notwithstanding, we did include references to a couple of hallmark works concerning colloidal systems, notably:

“Such gradual orientational ordering has indeed been observed in other colloidal systems that are vertically stacked (Ou *et al*) and may well exist here too.”

Dalmaschio et al., Impact of the colloidal state on the oriented attachment growth mechanism. *Nanoscale*, 2010, 2, 2336–2345

Also, there does not seem to be a quantitative kinetic plot such as an observable as a function of time. The work is not as quantitative as would be desirable.

We agree with the reviewer that a quantification of a temporal dependence could provide additional insights. Ideally, we would monitor individual nanocrystals or groupings thereof as a function of time and extract the translational and rotational velocities as they approach each other and make the jump to contact. That would require live imaging under ambient conditions (no cryogenic temperatures), as has been done for several inorganic systems (e.g. 10.1126/science.1219643). We have performed exploratory *in situ* TEM experiments using dedicated specimen holders to hold the protein molecules in an aqueous state while imaging, but these experiments have been plagued with radiation damage effects and do not reach the required resolution to be of service here.

Rather than following a single nucleation event as a function of time, we could adopt another approach where we extract quantitative data at the ensemble level. That strategy unfortunately suffers from one major limitation because nucleation is an ongoing process. In samples that were plunge-frozen at different points in time, we observe a broad range of crystalline states (nanocrystals, docked nanocrystals, mesocrystals, microcrystals, ...) that likely correspond to different points of the reaction coordinate. The relevant measurement of time would be the delay after primary nucleation has taken place, but that metric cannot be extracted from our images.

Having said that, we did perform a roughness analysis of the crystal facets, where we quantified the number of crystals that have a rough perimeter, and determined the Wenzel roughness parameter (see below).

Formal issues should be checked, such as language (e.g. line 195 "it's" should be "its") and inconsistencies in the references (including the authors' own names).

We thank the reviewer for the detailed revision of our manuscript, which has now been carefully checked for formal errors and other inconsistencies.

Introduction
line 52

"As it stands, two-step nucleation has emerged as the consensus in the field, but this fact remains difficult to reconcile with several issues." Are the authors implying that it is really _consensus_ for every type of system in this field ?

No, that is not the message we were trying to convey. We were trying to say that two-step nucleation is the most well-known and popular model. In many review articles, two-step nucleation is often presented as the de facto or most likely model to explain protein nucleation. Our current (and previous) observations for GI can be classified as being "non-classical", but are still very different from the "two-step" definition. To avoid any further confusion, we have made minor changes to the highlighted sentence:

"As it stands, two-step nucleation has emerged as the dominant model in the field, but that prepossession is unfounded for several reasons."

line 87

"We also point out that the facets, although rough on some occasions, are surprisingly smooth, suggesting that they represent a Wulff shape that emerges out of the anisotropy in the surface tension." It would improve the quality of the manuscript substantially if the authors could quantify this (how often is "some occasions" ? how rough ?)

In order to make such quantitative statements we have collected additional experimental data. Although we are working at high driving forces, the nucleation of these protein crystals is still quite a rare process. It can require many hours of cryoEM imaging to find and image only a handful GI nanocrystals. In the end, we have collected data on 118 crystalline particles. In this group, we do not take into account mesocrystalline arrays – we consider these to be formed in later stages of the assembly process and could therefore have been subject to different coarsening mechanisms. These particles are divided in two groups, namely (i) 57 isolated nanocrystals and (ii) 61 oriented attached crystals. 8 out of 41 nanocrystals have one or more rough facets, whereas 11 out of 36 OA crystals have one or more rough facets. As a measure of roughness, we have determined the "Wenzel roughness" (R_w) by calculating the ratio of the circumference of the nanocrystals and the theoretical, geometric circumference of a similarly sized crystal with smooth facets (Wenzel, 1936). For crystals with perfectly smooth facets, R_w is expected to be 1.0. For isolated nanocrystals, we obtain an R_w of 1.1 ± 0.3 (stand. dev., $n=24$).

Dimensionality issues (2D vs 3D and the differences between these; consequences of these)

- is this actually a (quasi) 2D study ?

Some of the images indeed give the impression that the GI nanocrystals are 2D monolayers, but they are in fact 3D crystals (the crystal lattice parameters can be found in the SI). This we infer from several observations:

- (i) the moiré patterns on the nanocrystals vary between different crystallites (e.g. Fig.3a) or between different regions of the same crystal (e.g. Fig.1d, red area in comparison to neighboring region). Such variations in interlacing corresponds to differences in crystal thickness along the c-axis which runs parallel to the incoming electron beam;
- (ii) Next to the patterning, we also see clear differences in contrast between neighboring crystals. Those changes in contrast emanate from changes in the crystal thickness along the c-axis (see image below);

- (iii) We cannot, however, make an unambiguous determination of the crystal thickness using only these moiré patterns and/or changes in contrast. For this we would need to image nanocrystals that are oriented perpendicular (or at least tilted with respect) to the plane of imaging. Unfortunately, all crystallites that we imaged had become self-aligned with the imaging plane. Such preferential orientation in absence of an anchoring substrate is quite common in cryoEM experiments with proteins. The typical thickness of cryogenically frozen liquid films after blotting is in the range of 100-200nm. It is therefore not surprising given the typical crystal dimension (50nm-1 μ m) that they have become aligned in this way. Having said that, we have invested considerable time in imaging additional grids to find crystals with outlier orientations. Below we present a number of images where such side-view orientations have been captured. The number of molecular layers varies from 3 to 10, but only small crystallites will be able to adopt this orientation given the typical ice thickness. The lattice spacing along this orientation is 8.3 ± 0.1 nm ($n=10$). The c-axis spacing that is expected based on X-ray diffraction is 7.83nm (Supporting Table 2). This means that there is a 5.7% expansion along the c-axis with respect to the X-ray structure. To put this in perspective, we looked at the typical variation in the lattice cell parameters for the I222 space group of GI based on depositions in the protein databank (www.rcsb.org). Based on 90 entries for the I222 space group, we arrive at a lattice variability of 2.8% (standard deviation / mean). To account for the remaining difference, we hypothesize that these nano-crystallites may adopt a more condensed state as they grow to larger sizes.

- how relevant are (quasi)2D studies to draw conclusions on 3D protein crystals which are used for structure determination ?

(Quasi)-2D studies absolutely have merit: the nucleation and growth mechanisms of 2D systems are very similar to their 3D counterparts because they are driven by the same physical mechanisms. The biochemistry and the resulting interaction potentials between the molecules that drive the system towards auto-assembly will be analogous for the 2D and 3D case. In many cases, it is the reduction of the dimensionality and with it, the reduction in complexity, that allows one to identify the fundamental principles that govern the system. A beautiful example of how 2D studies can drive scientific progress are the first experimental

observations of two-step nucleation (10.1021/la990397o; 10.1103/PhysRevLett.102.198302): at the time, only vague models were available that described how two-step nucleation might take place, but these two experimental works succeeded in resolving the complete nucleation pathway by leveraging the benefits of working with a system constrained to two dimensions (similar studies on 3D systems were not yet feasible at the time due to technical limitations). Years later, two-step nucleation pathways of 3D colloidal systems were also resolved experimentally and revealed many common features with their 2D analogues.

- interface effects (or from the limited reaction volume compared to "normal" bulk crystallization) ?

This is a valid question, but we believe that this is not an issue here. The reaction volume that we typically use is 50 μ l, i.e. sufficiently large to rule out any volume limitations. From this we take a 3 μ l aliquot that is applied onto a Cu-mesh grid. The excess liquid is immediately removed by applying pressure with blotting paper and the grid is rapidly driven into liquid ethane to cryogenically freeze the sample contents. The entire process takes only a matter of seconds. As a reference, the images presented in the manuscript correspond to incubation times (in the 50 μ l reaction volume) of 1min40s to 5min, i.e. one to two orders longer than the grid preparation time. We are therefore convinced that the kinetically frozen state of the system as we image it via cryoEM is representative to the processes that occur in the liquid bulk. We will point out that the most likely source of discrepancies between the imaged state and the state in the bulk is the blotting process which will (i) bias the EM sample towards smaller particles (as discussed above), and (ii) which will create solutal flows that will introduce additional shear forces. Having said that, we do see a clear temporal trend that starts from the emergence of isolated smaller crystals, to larger crystals with distinct grain boundaries and ultimately to mesocrystals. Such a trend can only have emerged in the 50 μ l reaction volume and is not an artefact of the cryoEM process.

- the nanocrystal monolayers are apparently aligned perfectly flat at the surface, none is tilted ? How about other orientations ? Can they be excluded ? Is this representative of bulk crystallization ?

Here we refer to our answer on the question regarding dimensionality. To summarize: Not all crystals are perfectly flat, we do find tilted specimens after collecting additional data. To reiterate further, these crystals are NOT monolayers, they are layered 3D crystals. Moreover, we have an indication that OA may also be directing the merger of nanocrystals along the direction of the c-axis. This is illustrated in the image below where we show a "side-view" image of a mosaic crystal. The annotated image on the right tentatively illustrates the various blocks within the larger assembly. The solvent gaps (arrows) that exist between different regions are of interest here. It is unlikely that such voids are created via traditional growth methods that are driven by monomer attachment processes. Rather, they are compatible with a mechanism of nanocrystal coalescence, suggesting that OA is not only limited to "in-plane" docking of nanocrystals.

- the behavior in the z dimension remains unclear.

Is there a "z-stacking" as observed for other pre-oriented building blocks (e.g. Ou et al., 2020, Kinetic pathways of crystallization at the nanoscale)?

No, we did not observe any significant z-stacking events, certainly not to the extent as is the case for Ou et al 2020.

Can the monolayers diffuse on top of each other? It seems that the monolayers merge first before a new layer grows on top. How does the new monolayer then grow? Is it a huge monolayer or several smaller ones diffusing on top? Or do only monomers attach in z-direction?

We believe that both processes occur concurrently, i.e. we have indications that nanocrystals can fuse together directly which would indeed entail a process of translational and rotational diffusion towards mutual registry. The image in Fig.3c illustrates a situation where such a process has either failed or has not yet reached completion. Alternatively, we also have indications that more classical layer generation mechanisms could be at play here. This is particularly apparent in Figure 1d. The area enclosed in red has a different lattice pattern than the surrounding area, which we attribute to differences in Z-height. That observation is compatible with the formation of new molecular layers on existing crystalline surfaces by means of 2D nucleation. That mechanism entails the formation of a small molecular island that grows laterally until it fully covers the underlying layer.

Reviewer #3 (Remarks to the Author):

This paper describes the nucleation behaviour of the protein glucose isomerase in the high rate regime where interactions between the formed particles cannot be ignored. The used technique, CryoEM, allows to determine molecular-scale time-stills during the nucleation process upon cryogenic freezing. The authors find that, rather than liquid-like, the formed

nanoparticles have a crystalline nature showing a structure of the macroscopic crystals. This thus seems to follow a classical route, as has been shown for other protein systems. However and moreover, these nanoparticles undergo oriented attachment (OA) among each other to form composite structures with grain boundaries and even perfectly aligned constructs. It is the first time that OA, a mechanism identified for inorganic nanocrystals, is observed for a protein system. The authors speculate that this mechanism is a universal mechanism for proteins, given the right fast nucleation rate conditions. Indeed they show with

this additional mechanism, that proteins can nucleate through diverse routes.

The paper is very convincing and the level of detail from the CryoEM technique is amazing. The paper is of very high importance to the community and outside it as it counters the two-step nucleation mechanism consensus that emerged in the community the last few years. The realization that each protein system picks, depending on the prevailing conditions, the least rate limiting route is interesting and leaves the question to what parameters determine that such a route is the fastest. It is therefore certainly a paper which I feel will have a high impact on the thinking in the field.

This realization of diverse routes available to the protein to nucleate would be appealing to show in a schematic.

Inspired by this suggestion, and the remark of reviewer 1 regarding the scheme in Figure 4, we have made a new schematic that summarizes the current state-of-the-art of protein nucleation.

The CryoEM strongly relies on the instant cryo-freezing by plunging the sample in liquid ethane. The reader might not be up to date with this technique. It would be good to explain the technique, either in the methods section or in the supplementary information, directed towards, on the one hand, the influence of the fast but not instantaneous heat transfer during plunging and, on the other hand, the effect of the measurement itself on the quality of the frozen sample.

We thank the reviewer for this excellent suggestion. We have added a more detailed description of cryoEM sample preparation to the methods section.

“Sample preparation was performed using an automated vitrification robot (FEI Vitrobot Mark III) for plunging in liquid ethane cooled to liquid nitrogen temperature (97K). All TEM grids were surface plasma treated for 40 seconds using a Cressington 208 carbon coater immediately before use. Three microliters of mother liquor solution was applied to a freshly glow-discharged grid in the humidity and temperature-controlled chamber of the Vitrobot. After 3 seconds of double-sided blotting, the grid was plunged into liquid ethane and vitrified at a cooling rate of > 104 K/s so that the sample is cryogenically preserved in amorphous ice. We chose t_0 as the moment where we induce supersaturation with respect to the crystalline phase (i.e. mixing of the protein with the precipitant solution). The exact time point t of the samples as indicated in the main text is defined as the moment (after blotting excess liquid) when the EM grid is plunged into the liquid ethane (t_{plunge}): $t = t_{plunge} - t_0$.

Imaging was performed in low-dose mode, a standard imaging technique where the search is performed at low magnification and focusing is done away from the area of

interest. The latter is only exposed at the final magnification when the image is taken to minimize radiation damage. For our samples, we did not exceed a total dose of 50 electrons/Å². Using this approach, we could confirm that the sample remained stable by collecting a second exposure which showed no obvious signs of radiation damage.”

The section "OA may be a universal mechanism for proteins" lacks clarity. For instance, the remark on a host of enigmatic experimental observations (Page 5, line 156) is seemingly followed by a seemingly unrelated general statement. It is unclear whether the "this" in line 158 relates to the observations or the general sentence following it. This sentence furthermore identifies a further nucleation mechanism where the 2-step mechanism is combined with non-oriented attachment, perhaps a mechanism that has to be highlighted earlier. Then a discussion follows on the intermediate species that may pre-exist in the used solutions, falling out of the blue sky. A reference to the reported seeding behaviour (line 171) is missing and it is actually unclear exactly what is meant with the term. Then, without further proof it is stated that the stock solution is slightly supersaturated (or is the red * in figure 5b indicating the solution conditions used?). It seems using supersaturated stock solution may cloud the interpretation of the observations as pre-existing clusters influence the nucleation behaviour. A growth rate is estimated (line 174) but it is unclear of what.

We agree with reviewer 3 that more clarity is needed to make the section more intelligible. For that, we would need to make it considerably longer to properly explain the experiments and associated control experiments that were performed. We fear that such an extension of this section would dilute or even convolute the main message of the manuscript. We have now removed this section in the revised version of the manuscript. If the editor and the reviewers do not agree with this decision, we will gladly incorporate an extended version of *OA may be a universal mechanism for proteins* into the text.

The authors clearly point out that OA seem rather unlikely as it needs the nanoparticles to experience a torque on a relatively large distance guiding the particles towards lattice alignment. They suggest that local water structure could play a role in this long range particle steering. Can the authors be a bit more specific?

For many systems, the primary cause of the long-range steering effect is electrostatic in nature. With a net dipole that is zero for a glucose isomerase molecule, it seems unlikely that nanocrystals are guided towards mutual lattice registry by ‘following local electric field lines’. In the absence of other clear explanations, some authors (such as Liu *et al*) have suggested that the solvent may somehow ‘communicate’ the orientation of a nanocrystal to its approaching neighbors. We should point out that Liu *et al* also write in very preliminary and succinct terms on this matter. Our very naïve working hypothesis is that the hydration shell which envelops the protein molecules, and by extension the nanocrystals, in some way mediates the contact between two crystals, well before any contact between two protein molecules is made. We speculate that the merging of the structured water layers contributes to the torque that steers the OA process. By now it is likely clear to the reviewer that we are in very speculative territory. In order not to mislead or misinform the future readers of this work, we decided to keep our comments on the solvent structure to a minimum.

The word "fact" in page 2 line 52 seems misplaced as after it reasons are given for the 2 step nucleation consensus not being a fact.

Thank you for pointing out that inconsistency. We have rephrased the sentence in the hopes of bringing more clarity:

“As it stands, two-step nucleation has emerged as the dominant model in the field, but that prepossession is unfounded for several reasons.”

Page 3, line 79/80: high nucleation rates: how high? Compared to what?

As we do not have clear quantitative data on the nucleation rate, we suggest replacing that statement with a more qualitative one. What we meant to say is that R387A GI forms hundreds of microcrystals in a volume of 10 μ l that are visible using a conventional light microscope within minutes after installment of supersaturation. Most proteins, however, take days or even months to nucleate into 3D crystals.

“In our optimized conditions, H32 forms hundreds of microcrystals in a mother liquor volume of 10 μ l within a timeframe of minutes. Such rapid nucleation increases the probability for nanocrystal interactions to occur in solution”

Page 3 line 90 and further, figure 1. The times in text and figure are not consistent.

We thank the reviewer for spotting this. We have corrected the errors in the text.

The last sentence of the introduction states OA to be a separate phenomenon from primary nucleation. This seems not consistent with the rest of the paper where it seems to be included in the nucleation process.

We agree with the reviewer that this sentence is not consistent with the rest of the paper. It has now been changed to:

“These observations highlight the underappreciated role of the interaction of nanoparticles, and the impact of OA, on nucleation process of protein crystals.”

Page 5 line 147: there is no article reference listed for the attractive force between nanoparticles in the presence of polymers.

Appropriate references have now been inserted.

What is actually the effect of PEG on the solubility of the protein?

The solubility of GI decreases exponentially as a function of PEG concentration (<https://pubs.acs.org/doi/abs/10.1021/cg800756h>).

Would the occurrence of OA then be a complex function of the effect of the used PEG chain length PEG on the solubility and of the induced attraction due to the kind of PEG used?

We do not have quantitative data on the effect of the PEG chain length on the solubility of GI, but for most proteins the solubility will tend to decrease if the chain length is increased while keeping the concentration constant. That effect can be explained by the depletion force

that is induced by the presence of the random polymers in the crystallization mixture. The length of the PEG chain will determine the length-scale of the depletion attractive force, and the concentration will determine the magnitude of the force. So, yes, the induced attraction will be a function of the type of PEG that is used. Having said that, it is not entirely clear how, and indeed if PEG contributes to the mechanism of OA other than in its capacity to lower the solubility. That is not a trivial question to answer and would constitute an entire new study.

The numbering of the figures is wrong at some places, for instance page 6, line 167, fig5b.

The numbering and other formal details have been carefully checked and corrected where necessary.

REVIEWER COMMENTS

Reviewer #1 (Remarks to the Author):

My concerns were addressed in part, and there was a reasonable explanation why some of them cannot be addressed (technical reasons). I have no further input, and the paper can be published.

Reviewer #2 (Remarks to the Author):

The authors wrote a lengthy response, but a number of points remain unclear.

re issue of quantitative kinetic plot

The response is rather lengthy

but in the end the requested information is not supplied

Unfortunately, other responses are in a similar style.

re line 87

It does not become clear from the response to the comment

how this entered the manuscript

re 2D vs 3D

It does not become clear from the response to the comment

how this entered the manuscript

The same applies to various other questions, including interface issues.

Since these questions will also be raised by some readers, the explanation, and preferably the additional information, should be supplied in the manuscript.

Lastly, I am irritated that one author was taken out, and I did not find a single line of explanation, which I would consider appropriate.

Reviewer #3 (Remarks to the Author):

The authors have dealt adequately with the comments from the reviewers and I am happy for the journal to accept as is.

Response to the Reviewers:

Reviewer #1 (Remarks to the Author):

My concerns were addressed in part, and there was a reasonable explanation why some of them cannot be addressed (technical reasons). I have no further input, and the paper can be published.

Reviewer #2 (Remarks to the Author):

The authors wrote a lengthy response, but a number of points remain unclear.

Our apologies for making the initial answers not clear enough. We hereby provide an updated manuscript that highlights the specific textual adaptations in response to some of the issues that were raised but remained unclear in the previous version. We employ a color-coded system of annotations that connects changes that were made to specific questions.

re issue of quantitative kinetic plot

The response is rather lengthy

but in the end the requested information is not supplied

Concerning the request for a quantitative kinetic plot, we unfortunately cannot provide such information at this point due to the experimental restraints of our setup. In brief, using cryoEM it is impossible to produce meaningful quantitative kinetic data, and at present we do not have the capability to perform liquid-cell TEM on our system. In fact, prior to the start of this study we performed exploratory liquid-cell TEM experiments in collaboration with Yuki Kimura (Hokkaido university), yielding only low resolution images (see image below) triggering us to use adopt the cryoEM methodology as used in the manuscript. In addition, protein molecules are highly sensitive to electron beam damage, making it uncertain that one would even be able to track the same sample as a function of time without inducing significant artefacts.

Fig: P222 GI crystals formed *in situ* from a 35 mg/ml GI, 1.4M ammonium sulfate solution imaged using (left panel) a microwell chip placed in a Poseidon TEM holder (Protochips Inc.,

Morrisville, NC); (right panel) conventional cryo-TEM image of plunge-frozen P222 GI crystals.

Unfortunately, other responses are in a similar style.

re line 87

It does not become clear from the response to the comment how this entered the manuscript

We refer to the newly revised and annotated manuscript.

re 2D vs 3D

It does not become clear from the response to the comment how this entered the manuscript

We refer to the newly revised and annotated manuscript.

The same applies to various other questions, including interface issues.

We apologize for not incorporating our answer into the main text. Please see the highlighted sections in the newly revised manuscript.

Since these questions will also be raised by some readers, the explanation, and preferably the additional information, should be supplied in the manuscript.

Lastly, I am irritated that one author was taken out, and I did not find a single line of explanation, which I would consider appropriate.

We apologize for not pointing this out in our response letter. Guy Schoehn was removed from the author list upon his own request as a section (“OA may be a universal mechanisms for proteins”) in the first submission has been removed in the resubmission to be included in another manuscript, because its pertinence to the present manuscript is peripheral. It was our careless omission that the acknowledgement was not updated. This omission has been fixed in the present resubmission.

Reviewer #3 (Remarks to the Author):

The authors have dealt adequately with the comments from the reviewers and I am happy for the journal to accept as is.

REVIEWERS' COMMENTS

Reviewer #2 (Remarks to the Author):

I consider the manuscript now acceptable for publication.